# Effect of Chitosan Deacetylation on Its Affinity to Type III Collagen: A Molecular Dynamics Study

**DOI:** 10.3390/ma15020463

**Published:** 2022-01-08

**Authors:** Piotr Bełdowski, Maciej Przybyłek, Alina Sionkowska, Piotr Cysewski, Magdalena Gadomska, Katarzyna Musiał, Adam Gadomski

**Affiliations:** 1Institute of Mathematics & Physics, Bydgoszcz University of Science & Technology, 85-796 Bydgoszcz, Poland; agad@pbs.edu.pl; 2Department of Physical Chemistry, Pharmacy Faculty, Collegium Medicum of Bydgoszcz, Nicolaus Copernicus University in Toruń, Kurpińskiego 5, 85-950 Bydgoszcz, Poland; m.przybylek@cm.umk.pl (M.P.); Piotr.Cysewski@cm.umk.pl (P.C.); 3Department of Biomaterials and Cosmetics Chemistry, Faculty of Chemistry, Nicolaus Copernicus University in Toruń, Gagarin 7, 87-100 Toruń, Poland; alinas@umk.pl (A.S.); 291013@stud.umk.pl (M.G.); musialk.97@gmail.com (K.M.)

**Keywords:** chitosan, hyaluronic acid, collagen, bio-polymers, molecular docking, intermolecular interactions

## Abstract

The ability to form strong intermolecular interactions by linear glucosamine polysaccharides with collagen is strictly related to their nonlinear dynamic behavior and hence bio-lubricating features. Type III collagen plays a crucial role in tissue regeneration, and its presence in the articular cartilage affects its bio-technical features. In this study, the molecular dynamics methodology was applied to evaluate the effect of deacetylation degree on the chitosan affinity to type III collagen. The computational procedure employed docking and geometry optimizations of different chitosan structures characterized by randomly distributed deacetylated groups. The eight different degrees of deacetylation from 12.5% to 100% were taken into account. We found an increasing linear trend (R^2^ = 0.97) between deacetylation degree and the collagen–chitosan interaction energy. This can be explained by replacing weak hydrophobic contacts with more stable hydrogen bonds involving amino groups in *N*-deacetylated chitosan moieties. In this study, the properties of chitosan were compared with hyaluronic acid, which is a natural component of synovial fluid and cartilage. As we found, when the degree of deacetylation of chitosan was greater than 0.4, it exhibited a higher affinity for collagen than in the case of hyaluronic acid.

## 1. Introduction

Many of the synovial joints’ diseases, such as degeneration and inflammation, are associated with the deficiency of essential components forming cartilage and synovial fluid. These compounds are often classified as extracellular matrix molecules. The essential proteins belonging to this class, namely collagens, play the chief structural framework role in many tissues, including cartilage, tendons, skin, blood vessels, and bones. Collagen is a supramolecular system characterized by a triple helical structure. The collagen triple helices form the “building blocks” of tissues, namely fibrils and fibers. Furthermore, the collagen nanofibrils can be used for obtaining various functional materials (e.g., gels, films, and microgels) [1]. Interestingly, the molecular dynamics (MD) calculations results reported by Mathavi et al. (2019) [2] showed that the collagen self-assemblies are stabilized by the interchain water-mediated hydrogen bonds (H-bonds). The capability of forming strong intermolecular interactions determines specific features of collagen, such as bio-compatibility, low antigenic activity, bio-degradability, and tissue regeneration abilities [3,4,5,6,7,8].

The collagen properties can be modified by using various additives. Many studies, dealing with pharmaceutical, cosmetic, and tissue engineering applications of collagen materials modified by various low- and high-molecular components, have appeared recently. For special attention deserve collagen blends containing phenolic compounds [9,10,11,12,13,14,15], polysaccharides, and their analogs, including carboxymethylcellulose, hyaluronic acid (HA), and chitosan [16,17,18,19,20,21,22,23,24,25,26,27,28,29,30,31]. Probably one of the most important modifications is associated with mechanical properties’ improvement. For instance, the collagen–ferulic acid films were found to exhibit a higher maximum tensile strength, elongation at break, and so high-strength Young’s modulus than in the case of pure collagen films [9]. These features are important for obtaining collagen scaffolds characterized by improved mechanical durability [7,32,33]. On the one hand, chitosan is probably, the most popular polymeric additive, which was widely used to prepare collagen scaffolds characterized by enhanced mechanical stability [21,22,23]. On the other hand, HA-collagen systems are generally characterized by low mechanical durability [16]. Both bio-polymers, HA and chitosan, are characterized by good miscibility with collagen [20], making them a promising bio-materials component. The former is a polysaccharide analog belonging to the non-sulfated glycosaminoglycan class consisting of D-glucuronic acid and *N*-acetyl-d-glucosamine moieties. The HA-collagen systems are inherently associated with diseases and connective tissue regeneration of synovial joints. The anti-osteoarthritis activity of HA and sodium hyaluronate is mainly associated with the replenishment of the aging-induced deficiency of this polysaccharide in articular cartilage and synovial fluid (viscosupplementation) [34]. However, this simple application is not the only role that HA can play. The outstanding properties of this bio-lubricant are essential from the tissue protection and regeneration perspective. The tissues repair processes are supported by HA-induced mechanisms such as inflammation and cellular migration [35,36]. Other vital features of HA, namely moisturizing properties, reactive oxygen species quenching, and its bio-lubricative affinity to lipids, are beneficial for preventing skin aging and/or degradation [37,38,39].

A quite similar structure to HA characterizes chitosan. This polymer consists of D-glucosamine and N-acetylglucosamine moieties. Similarly, as in HA, the saccharide units are jointed with β-(1→4) glycosidic bonds. However, in the case of HA, there are additional β-(1→3) glycosidic bonds between D-glucuronic acid and N-acetyl-D-glucosamine moieties. Chitosan is formed by partial deacetylation of chitin, and its properties are strictly associated with the degree of deacetylation, which usually ranges from 50–100% [40]. The acetyl groups enhance many properties, including viscosity, solubility, hydrophilicity, hygroscopicity, bio-compatibility, analgesic activity, mucoadhesion abilities, and membrane permeability [30,40,41]. It is worth stressing that a number of studies have shown significantly better properties of chitosan characterized by a high or ultrahigh deacetylation degree for bio-material engineering [42,43,44]. According to Hsu et al. (2004) [42], who studied the chitosan/alginate blends, DD affects important mechanical properties such as tensile strength, which can be explained by the higher polarity of deacetylated chitosan, leading to a more compacted structure of the complex. The intriguing properties of chitosan make it a promising HA substitute in tissue regeneration. Indeed, the ability to modulate chitosan properties by changing the deacetylation degree (DD) enabled the design of collagen composites to reconstruct various tissues such as skin, adipose tissue, and cartilage [23,30,45,46].

There are various bio-medical, tissue engineering, and cosmetic applications of collagen blends, including chitosan hydrogels, films and nanofibers. Some interesting examples are wound healing [47,48,49,50,51], artificial skin [52,53,54] and rheological modifiers of personal care products [55] and locomotor system disease treatment [56]. The latter example deserves special attention due to the global prevalence of joint diseases in recent years [57,58]. The beneficial effects of collagen blends for joint health are associated with healing natural tribological systems, namely synovial fluid and articular cartilage. Moreover, the hydrophilicity of hydrogels is closely related to gel-like and hydration involving propensity at the expense of their sol-type character, especially when being affected by slight temperature changes experienced around the physiological reference point [59,60]. Therefore, the characterization of intermolecular interactions, including the analysis of hydrophilic regions capable of hydrogen bond formation and hydrophobic regions responsible for van der Waals forces and dispersion (colloid type) interactions, is useful in the design of new bio-materials. It can also support a view of identifying the facilitated lubrication in bio-systems with a nanoscale-oriented undocking effect, whereas a complete docking ought to be reserved for extremely obstructed lubrication [61,62]. Furthermore, as it was established, the molecular dynamics techniques can be useful in explaining the effect of hydration on the components of the synovial fluid lubricating properties [61,63]. In our previous work [64], the affinity and intermolecular interactions between synovial fluid components, namely human serum albumin and hyaluronic acid and Ca^2+^, Mg^2+^, Na^+^ cations, were described and the results obtained were consistent with the experimentally-assessed observations reported in the literature.

When considering collagen-based bio-material features, it is important to emphasize that there are 29 different types of collagen [10]. For instance, in the articular cartilage, at least five types of collagen can be distinguished, and type II is the most abundant [65]. On the other hand, tendon connective tissue consists mainly of collagen type I [66]. In our study, type III collagen was taken into account since it plays an important role in tissue regeneration [67], which seems interesting from the regenerative medicine viewpoint. Noteworthy, type III collagen appearance in cartilages is often associated with osteoarthritis [68]. However, type III collagen was also found in healthy adult hip articular cartilage [69]. Notably, type III is deposited on type II collagen, which plays the framework role [69]. According to the recent study of Wang et al. (2020) [70], the presence of type III collagen significantly affects the bio-mechanical features of articular cartilage since it plays a crucial role in maintaining the appropriate shape of fibrils [70]. Noteworthy, various studies showed that cartilage injections with chitosan hydrogels could be successfully applied for the osteoarthritis treatment leading to a significant reduction of pain and local inflammation [71,72,73,74].

The main goal of this study is to describe the deacetylation effect on intermolecular interactions in human type III collagen–chitosan systems using molecular docking followed by molecular dynamics simulations. The intermolecular interactions analysis will help explain the affinity of chitosan to collagen found in osteoarthritic cartilage. Furthermore, the obtained results will be compared with HA, which naturally occurs together with collagen in human tissues. Such comparison allows for evaluating the usability of chitosan in the context of its application in tissue engineering and bio-materials’ design.

## 2. Methods

We used the collagen-like structure (PDB code 1BKV) from the protein data bank (PDB), with the structure T3-785, (Pro-Hyp-Gly)3-Ile-Thr-Gly-Ala-Arg-Gly-Leu -Ala-Gly-Pro-Hyp-Gly-(Pro-Hyp-Gly)3, which includes the 12-residue amino acid sequence 785–796 of human type III collagen and is capped with (Pro-Hyp-Gly)3 triplets to enhance folding and triple-helical stability. The collagen III and collagen I form characteristic 670 Å fibrils in various tissues, including skin and blood vessels [75]. Since this study’s main goal is associated with the chitosan deacetylation and its consequences on the affinity toward the collagen, the chitosan structures characterized by a different deacetylation degree (DD) were constructed. The general chitosan deacetylation scheme can be visualized in Figure 1. First, chitin of a molecular weight of 3 kDa from the PubChem database was modified to obtain chitosan of 8 different degrees of deacetylation (DD) from 12.5% to 100%. Then, ten variants of every DD were prepared to obtain more structures with randomly distributed deacetylated groups. Next, HA structure was obtained from the PubChem database and extended to a molar mass of ~3 kDa. Since no modifications were required, only one HA chain variant was used for further investigation. Finally, both structures were minimized to obtain optimal geometry. We docked complexes using the VINA method [76] with default parameters and point charges initially assigned to the AMBER14 force field [77] for collagen and the GLYCAM06 force field [78] for Chitosan and HA. It was then modified in order to be more similar to Gasteiger charges exhibiting lower polarity applied for AutoDock scoring function optimization. These computations were performed using YASARA software [79,80]. The best hit of 50 runs with −10 kcal/mol free binding energy was selected as distinct complexes. As a result, we obtained 150–180 complexes (15–20 for every variant of a given deacylated molecule). Chitosan of DD = 100% and HA have only one variant; therefore, those formed only ~20 complexes each. In total, we obtained ~1200 complexes that were further used.

Next, all the structures were simulated in an aqueous solution with molecular dynamics simulation. The protocol covered the hydrogen bonds optimization [81] to obtain suitable peptide chain protonation microstates at pH = 7.4 [81]. Then, after the annealing simulations and the steepest gradient achievement, the MD computations were performed for one ns applying the AMBER14 force field for the collagen, GLYCAM06 for HA, chitin, and chitosan, and TIP3P for water [82]. The default AMBER settings, namely 10 Å-threshold was used for van der Waals interactions, no threshold was utilized at the electrostatic forces identification step (Particle Mesh Ewald algorithm, [83]). Next, the motions equations integration was performed using 1.25 fs and 2.5 fs multiple time steps in case of bonded interactions and non-bonded interactions, respectively (*T* = 310 K *p* = 1 atm, NPT ensemble, default protocol [83]). After the solute RMSD (root mean square deviation/displacement) analysis, 100 points from the 0.9–1 ns range were selected and used for binding energy and other parameters determination.

### 2.1. Binding Energy Computation

The key parameter analyzed in this paper was binding energy denoted by, $E_{bind}$. In this study, $E_{bind}$ was determined using the abovementioned MD methodology and software according to the definition expressed by Equation (1). If the $E_{bind}$ is negative, the high affinity characterizes the ligand to protein [84].
(1)$$E_{bind}=E_{pot-comp}+E_{sol-comp}-\left(E_{pot1}+E_{pot2}+E_{sol1}+E_{sol2}\right)$$
where ***E*_*pot*1_** and ***E*_*pot*2_** denote target protein and ligand potential energy values, ***E*_*solv*1_** and ***E*_*solv*2_** stand for solvation energies of collagen and are considered polysaccharides (chitosan or hyaluronic acid), ***E*_*pot-comp*_** is the intermolecular potential energy of the complex, while ***E*_*sol-comp*_** denotes the complex solvation energy.

### 2.2. Hydrogen Bonding Definition

This study used the default YASARA settings and hydrogen bonding definition (i.e., the energy calculation protocol according to Equation (2) and 6.25 kJ/mol energy threshold [84]).
(2)$$E_{HB}=25\cdot \frac{2.6-max\left(Dis_{H-A,}2.1\right)}{0.5}\cdot Scale_{D-A-H}\cdot Scale_{H-A-X}$$

In Equation (1), the first scaling factor is significantly influenced by Donor–Acceptor– Hydrogen angle, while the second scaling factor can be determined from the Hydrogen–Acceptor–X angle, where the X symbol denotes the covalently bonded atom to the acceptor. The scaling factors’ values are in the range of 0 to 1.

### 2.3. Hydrophobic Interactions

The hydrophobic interactions were identified using the default YASARA settings [79]. According to these rules, the hydrophobic interactions are formed by carbon atoms with three/two hydrogen/carbon atoms attached and C-H moieties in aromatic carbon rings.

### 2.4. Ionic Interactions

The ionic interactions are defined as two atoms’ contacts being the distance between formal integers centers. Noteworthy, according to the YASARA approach, the distances corresponding to hydrogen bonds are subtracted. Direct ionic interactions found in collagen–chitosan complexes are formed by ARG and chitosan hydroxyl groups.

## 3. Results and Discussion

This study analyzed the intermolecular interactions between chitosan and collagen type III using molecular docking methodology followed by molecular dynamics simulations. Additionally, the affinity of chitosan to the target protein was compared with HA. The collagen type III secondary structure is characterized by the dense helical shape (homotrimer). The collagen helix is stabilized by the strong N-H(GLY) ⋅⋅⋅ O = C (Xaa) hydrogen bonds [85]. Due to this ordered and compact structure, the intermolecular interactions with other bio-polymers can be formed by the side groups in the peptide chain. The exemplary structures of complexes characterized by a different chitosan deacetylation degree (DD) were presented in Figure 2. As one can see, a completely deacetylated chitosan (DD = 100%, Figure 2c) exhibits a parallel orientation in relation to collagen contrary to hyaluronic acid and chitosan–collagen complexes characterized by lower DD values. Characteristic hydrogen bonds for low DD are formed mainly between the hydroxyl group and HYP/GLY/ARG. For higher DD more amino groups can form more H-bonds, primarily by GLY. The shortest distance between donor and acceptor formed by ARG is about 2 Å.

Interestingly, ARG moiety was added to cross-link collagen with chitosan [86]. This amino acid plays an essential role in collagen triple helix assembly [87]. Both ARG and HYP are considered the least destabilizing amino acids in a collagen helix’s Y and X positions, while GLY is a highly destabilizing residue [88].

The peptide molecule considered in this work consists of 34% of GLY, 24% of HYP, 23% of PRO, ~3% of ARG. The position of each amino acid is presented in Figure 3. As one can see, ARG lays in the middle of the collagen molecule, whereas HYP (as well as PRO) does not appear in that region. On the other hand, GLY lays everywhere throughout the structure. This result suggests that deacetylation opens more binding sites for chitosan as ARG is a less common amino acid in H-bond formation (this is also why there is an increase in H-bond formed with GLY).

It is worth noting that many physicochemical properties, including solubility, are proportional to the degree of deacetylation [40]. A simple linear relationship characterized by the good correlation quality (R^2^ = 0.97) can also be observed in the case of affinity (Figure 4a). Since the structural and energetical analysis performed in this study covered collagen complexes formed by both chitosan and HA, the properties of these macromolecules can be compared. It is expected that the high affinity of the polysaccharide to the protein will affect the mechanical properties of the collagen composite. Based on the analysis presented in Figure 4a, one can select the chitosan structures, which are supposed to be more compatible with collagen than HA. As it can be inferred in the case of DD > 40%, the affinity of chitosan to collagen is generally higher than in the case of hyaluronic acid, which is consistent with the higher mechanical stability of the systems formed by collagen and highly deacetylated chitosan [21,22,23]. Importantly, the study by Lewandowska et al. (2016) [20] showed that ternary collagen/hyaluronic acid/chitosan blends are characterized by higher tensile strength than binary collagen/hyaluronic acid mixtures. Noteworthy, the very high affinity of chitosan (DD = 85%) to collagen was confirmed by the miscibility analysis using viscosimetric measurements [89]. According to this study, chitosan is miscible with collagen in solid and liquid at 25 °C in all ratios. In the case of a liquid state, the miscibility parameter, Δ*b* for different chitosan–collagen blends, was in all cases higher than 0 and ranged from 0.0857–0.2543 (L/g) ^2^ [89].

As one can see from Figure 4b,d, there is a very weak correlation between the number of ionic interactions and DD (R^2^ = 0.22) and practically no correlation in the case of the number of hydrogen bonds vs. DD plot (R^2^ = 0.01). This is understandable as the number of intermolecular interactions is only a very rough measure of the stability of macromolecular complexes [90]. On the other hand, the stability of collagen self-assemblies is related to the number of intermolecular interactions, mainly hydrogen bonds and electrostatic interactions [10,91,92]. It is worth emphasizing that both interactions are characterized by the highest contribution to the binding energy. When the DD is higher than 0.3, the number of hydrogen bonds is, in general, slightly higher than in the case of HA. The hydrogen bonds distributions corresponding to each collagen amino acid are presented in Figure 5. As one can see, in all cases, HYP is forming the highest number of hydrogen bonds. The highest percentage (44%) was observed in the case of 12.5% DD. Noteworthy, the total number of hydrogen bonds for the chitosan structures characterized by low DD (≤0.25) is relatively small (Figure 4b). Interestingly, in the case of DD = 62.5%, a decrease in the H-bonds can be observed. The fully deacetylated chitosan (DD = 100%) is characterized by the lowest number of hydrogen bonds and the highest affinity to the peptide (Figure 4a). However, it should be taken into account that DD does not affect significantly the number of collagen–chitosan H-bonds. The relative difference between the highest and the lowest average number of H-bonds is c.a. 6%. Therefore, the number of hydrogen bonds should be analyzed critically, and the distribution of individual contacts is more important from the collagen–chitosan affinity perspective. In the case of fully deacetylated chitosan, the highest distribution of hydrogen bonds was formed by HYP (44%). It is worth mentioning that HYP plays a crucial role in collagen helix conformational stabilization [93]. Therefore, it should be expected that chitosan will destabilize the tertiary structure of collagen. Indeed, the collagen–chitosan wide-angle X-ray scattering (WAXS) analysis of the 0.84 and 3.5 nm^−1^ regions showed that helix structure is less pronounced with the increase in chitosan, as evidenced by the intensity reduction of the peak corresponding to the intermolecular interactions between collagen chains [94]. This is understandable since the formation of new intermolecular interactions of chitosan with the peptide chain disrupts the triple helix system.

As can be seen from Figure 5, the GLY and ARG hydrogen bonding contributions are clearly related to DD. The contribution of GLY significantly increases with the increase in DD. On the other hand, the contribution of H-bonds formed by ARG decreases. This is understandable since collagen type III has much more GLY than ARG residues.

The ionic interactions were found only in the case of chitosan–collagen complexes, which can be explained by the proton transfer between zwitterionic amino acid moieties and the amino group in chitosan. The highest number of ionic interactions and hydrogen bonds was observed for DD = 75.0%. Interestingly, the number of ionic interactions is the lowest for DD = 100%. Furthermore, a slightly ascending trend for DD < 37.5% can be distinguished. It can be concluded that when the DD > 50%, the number of ionic contacts dramatically increases. This is a quite surprising observation, which confirms the complex polyelectrolyte nature of collagen. Noteworthy, another important property, namely water solubility, which is closely related to macromolecules’ polarity and electrostatic features, was also found to be significantly enhanced when DD > 50% [95]. However, in the case of fully deacetylated chitosan, the strong H-bonds employing amino groups much more predominantly contribute to the complex stabilization, which is probably caused by the close parallel orientation of chitosan and collagen (Figure 2c), posed by the lack of sterically hindering acetyl groups. Such an efficient assembly of chitosan and collagen macromolecules occurs only in the case of DD = 100%, which explains the abrupt change observed in Figure 4d. Figure 6 shows the electrostatic potential map of selected structures. Both collagen and chitosan are characterized by scattered negatively charged regions. In the case of collagen, the negative charge is mainly located on oxygen atoms, while in the case of chitosan, in the case of all oxygen and nitrogen atoms, although in N-acetyl moiety, the negative charge is delocalized due to the resonance effect. The appearance of a positive charge in the case of both bio-polymers is generally associated with protonated amine groups, which play a quite important role in the cross-linking mechanism [86]. In Figure 7, the proton transfer between THR in collagen and amino group in chitosan was shown. As one can see, there is a characteristic positively charged small region in the center of the electrostatic potential map, corresponding to the protonated amino group of chitosan. Noteworthy, the appearance of both ionic intermolecular interactions involving -NH_3_^+^ and -COO^−^ groups was confirmed for gelatin–chitosan and collagen–chitosan blends using infrared spectroscopy [94,96].

Hydrophobic contacts energy is relatively weak with up to 2 kJ/mol strength. On the other hand, H-bond energy is between 6 and 25 kJ/mol; ionic interactions are in between the strength of those interactions but are much less common. Although H-bonds are the most important factor when describing the stability of collagen–chitosan complexes, the hydrophobic interactions also play an important role in complex stabilization. Interestingly, an excellent correlation (R^2^ = 0.98) between the hydrophobic interactions number and the deacetylation degree was observed (Figure 4c). This seems to be quite intuitive since the acetyl moiety can form weak interactions due to the presence of the methyl group. Finally, although the number of ionic interactions is small, it significantly increases for higher DD. This effect explains the decrease in ARG hydrogen bonds contribution, as the ionic interactions with collagen hydroxyl groups are more abundant.

A certain conceptual pathway to follow may also rely on rationalizing the assertion, which emphasizes the microscopic conditions of emerging the completely deacetylated chitosan (DD = 100%) as characterized by its parallel orientation in relation to collagen. This can be envisaged when invoking the overdamped stepper model by Martin Bier [62], in which a walk of one bio-polymer over the other involves docking and ratcheted diffusion of the corresponding chemical groups belonging to them. The model realization of such a two-step process, if extendable out of the motor proteins’ realm, relied on attachment (docking) and detachment (controlled by diffusion) of the chemical groups engaged in the overall process. This also supports qualitatively the detailed view of the partial vs. complete deacetylation of chitosan and collagen studied by the present paper.

## 4. Conclusions

Polymer–polymer interactions are essential in preparing new bio-materials based on bio-polymer blends. Our results show clearly that the increase in deacetylation degree of chitosan results in its higher affinity to type III collagen, being the consequence of the formation of more hydrogen bonds by GLY rather than by ARG found in a middle zone. Although the number of hydrogen bonds does not significantly change, their characteristic distribution in the case of highly deacetylated chitosan allows for the formation of more stable complexes. The crucial finding is the high linear correlation between binding energy and deacetylation. This result is consistent with the frequently observed linear trends of many important physicochemical properties of chitosan as a function of deacetylation degree. Short simulation time, the low molecular mass of chitosan and collagen is the limitation of the applied theoretical approach. However, our results can give an impulse to design the experiment to study the problem presented here. Therefore, it seems interesting to extend the research on the affinity of chitosan to other bio-polymers in the context of new bio-materials screening. It is worth mentioning that the presented results included the analysis of the affinity of hyaluronic acid to collagen, which is a kind of reference point. Therefore, it can be assumed that chitosan, showing a similar affinity to collagen as hyaluronic acid, will exhibit similar cross-linking properties, which is essential from the new materials design perspective. However, to obtain a better insight into the chitosan–collagen interactions, the research should be expanded to understand the influence of other factors, i.e., temperature, ionic composition, or pH, on collagen–chitosan complexes’ stability. Furthermore, it is interesting to compare the stability of the chitosan type III collagen complex with molecular systems formed by other collagen types.

## Figures and Tables

**Figure 1 materials-15-00463-f001:**
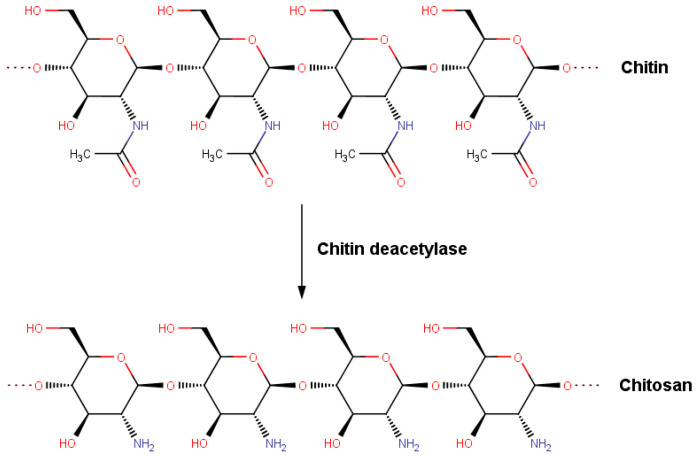
Deacetylation of chitin into chitosan.

**Figure 2 materials-15-00463-f002:**
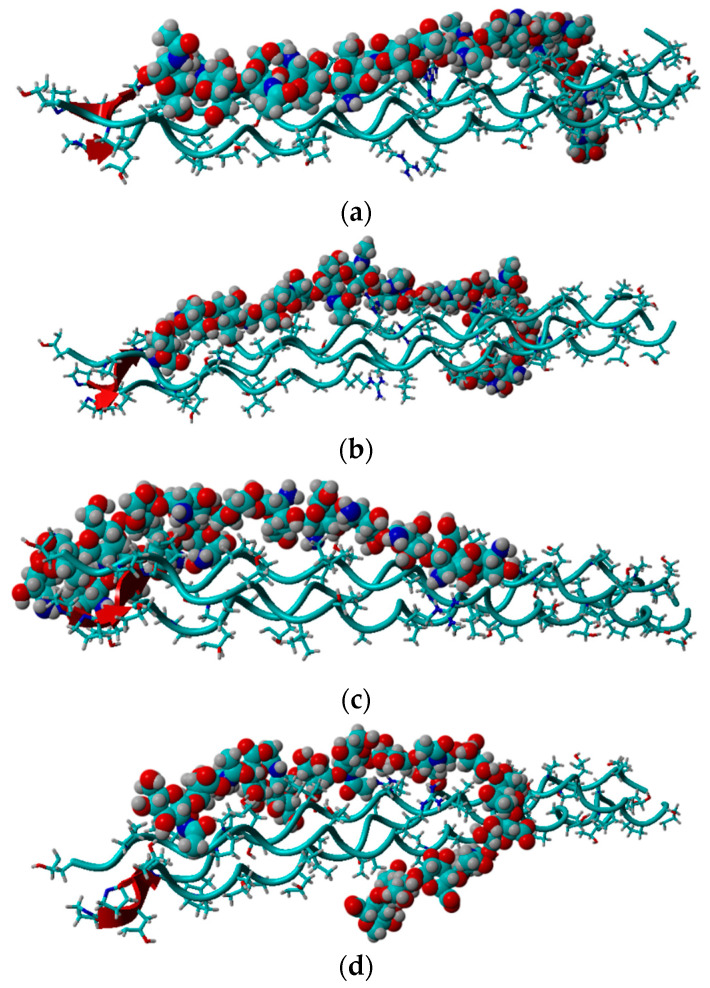
Snapshot from docking experiment showing optimized complexes for (**a**) DD = 12.5%, (**b**) DD = 50%, (**c**) DD = 100%, (**d**) HA.

**Figure 3 materials-15-00463-f003:**
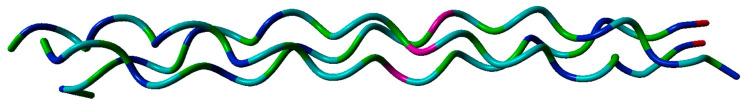
The optimized structure of collagen. ARG, GLY, and HYP amino acids are denoted by pink, dark blue, and green colors, respectively.

**Figure 4 materials-15-00463-f004:**
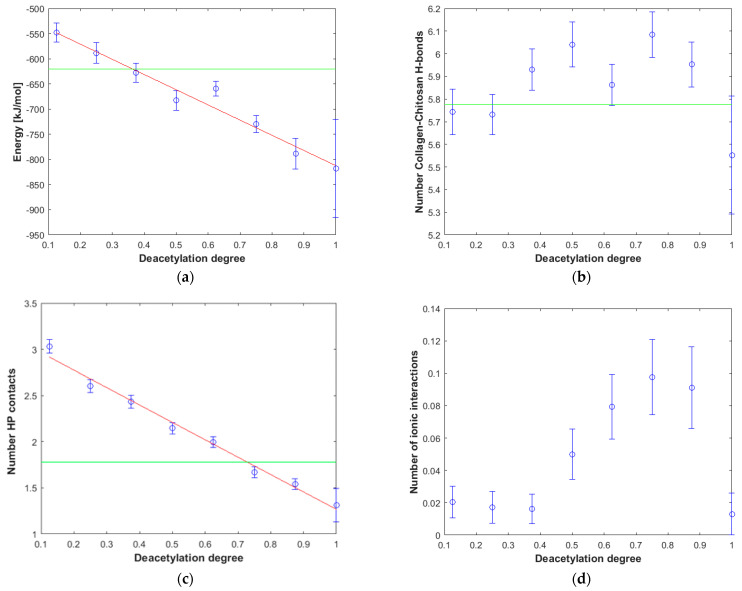
The influence of deacetylation on the stability and selected structural features of collagen–chitosan complexes. (**a**) Affinity energy vs. deacetylation degree (DD) of chitosan. (**b**) Number of intermolecular H-bonds vs. deacetylation degree (DD) of chitosan. (**c**) Number of intermolecular hydrophobic (HP) interactions vs. deacetylation degree (DD) of chitosan. (**d**) Number of intermolecular ionic interactions vs. deacetylation degree (DD) of chitosan. The green line represents the average affinity energy value determined for the collagen-HA system.

**Figure 5 materials-15-00463-f005:**
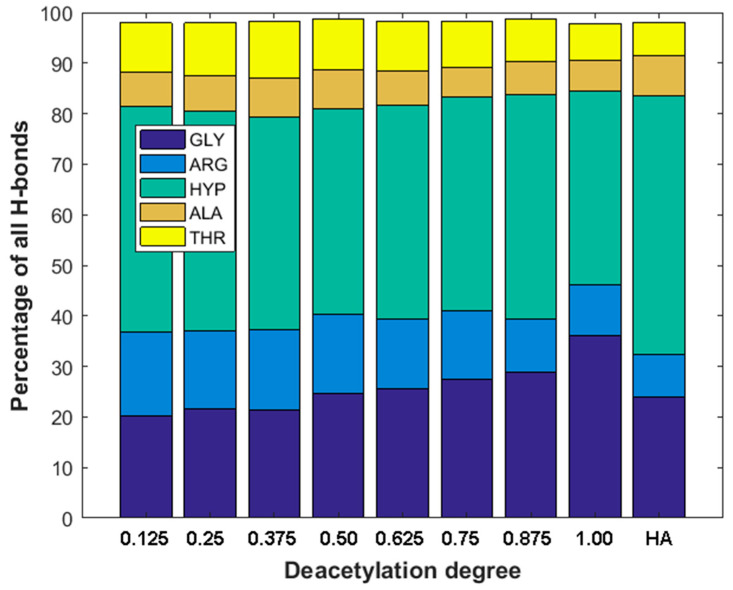
Distribution of H-bonds for most important amino acids vs. deacetylation degree (DD) of Chitosan and H-bonds distribution in hyaluronic acid–collagen complex (HA).

**Figure 6 materials-15-00463-f006:**
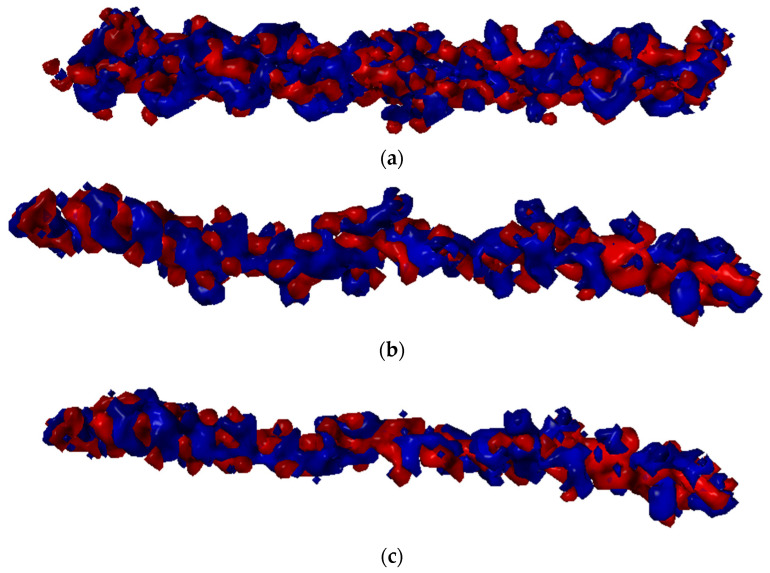
Electrostatic potential maps determined for selected structures of Collagen (**a**) and chitosan DD = 25% (**b**), DD = 50% (**c**), DD = 75% (**d**), and DD = 100% (**e**). Blue color stands for the positive charge, white red color denotes negative charge.

**Figure 7 materials-15-00463-f007:**
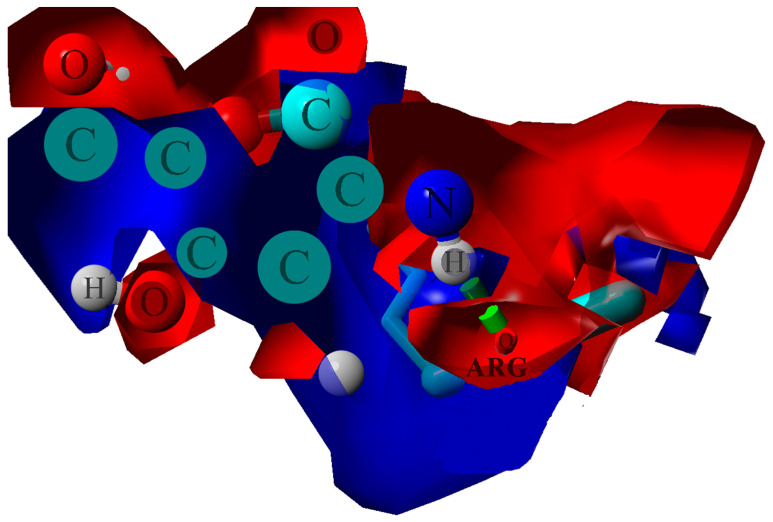
Electrostatic potential map determined for ARG–chitosan (DD = 87.5%) contact. The blue color stands for the positive charge, and the red denotes the negative charge. The green dotted line indicates H-bond.

## Data Availability

All data are available in the paper or upon request to the corresponding author.

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
