# Peer review of "Effect of Chitosan Deacetylation on Its Affinity to Type III Collagen: A Molecular Dynamics Study"

_materials, 2022, doi:10.3390/ma15020463_

Round 1
Reviewer 1 Report
1. In general, the document turns out to be relevant research, in fact, few publications stand out in this regard. However, the writing style of manuscript deserves a major revision, both grammatically and scientifically. 2. The abstract is poorly structured, they are a series of ideas without a common thread that allows us to understand the research carried out. The objective or methodology of the investigation is not detailed in abstract. The authors could highlight some values of the results obtained and thus provide greater value to the results obtained. 3. The introduction has relevant information, but the writing is deficient and causes you to move from one topic to another abruptly, that is, there is a lack of cohesion between the ideas in the paragraphs. 4. All abbreviations must be explained on their first appearance. 5. The results are good, but the explanation of these ends up lacking bibliographic support. Particularly from line 226 and up to line 259, there is not a single reference to support the claims made by the authors. 6. In the introduction, the medical importance of biomaterials is mentioned a lot, but the results do not mention anything about how the results obtained could represent that added value for biomedicine. 7. The conclusions are the authors' own, which are reached from the results. Reason why, they should not carry bibliographic citationsAuthor Response
First of all, I would like to thank the Reviewer for his valuable comments that have contributed to a significant improvement of our manuscript. Responses to all comments are listed below. We have also changed to the title to better describe the collagen type used in our study.
- In general, the document turns out to be relevant research, in fact, few publications stand out in this regard. However, the writing style of manuscript deserves a major revision, both grammatically and scientifically.
Thank you, we have improved the style and corrected grammar errors.
- The abstract is poorly structured, they are a series of ideas without a common thread that allows us to understand the research carried out. The objective or methodology of the investigation is not detailed in abstract. The authors could highlight some values of the results obtained and thus provide greater value to the results obtained.
Thank you for this comment. We improved the abstract.
- The introduction has relevant information, but the writing is deficient and causes you to move from one topic to another abruptly, that is, there is a lack of cohesion between the ideas in the paragraphs.
Thank you. We tried to improve the Introduction section and make it more consistent.
- All abbreviations must be explained on their first appearance.
Thank you, we corrected this.
- The results are good, but the explanation of these ends up lacking bibliographic support. Particularly from line 226 and up to line 259, there is not a single reference to support the claims made by the authors.
Thank you. We added some information from the literature to discussion.
- In the introduction, the medical importance of biomaterials is mentioned a lot, but the results do not mention anything about how the results obtained could represent that added value for biomedicine.
Thank you for this comment. We improved the Introduction section.
- The conclusions are the authors' own, which are reached from the results. Reason why, they should not carry bibliographic citations
Thank you. It was corrected.
Reviewer 2 Report
“In the manuscript “Effect of chitosan deacetylation on its affinity to collagen: molecular dynamics study” presents the molecular dynamics of chitosan deacetylation’s effect on affinity towards collagen. The Manuscript is planned and prepared very well.
the authors address the question what type of effects of the deacetylation has on the intermolecular interactions in collagen-chitosan systems using molecular docking followed by molecular dynamics simulations. The topic is relevant to the computational chemistry of the biomaterials. The flow of the manuscript is fine and understandable to the readers.
The presented analysis will create an interest of the readers in the affinity of chitosan towards the peptides and usability of chitosan in designing for tissue engineering and biomaterials. Authors address the question satisfactorily, considering their approach in the presented methodology and computational model.
However, the conclusion can be improved a bit more to the specific to the question in rather generalizing.
Recommended for acceptance.”
Author Response
First of all, I would like to thank the Reviewer for his valuable comments that have contributed to a significant improvement of our manuscript. Responses to all comments are listed below. We have also changed the title to better describe the collagen type used in our study.
In the manuscript “Effect of chitosan deacetylation on its affinity to collagen: molecular dynamics study” presents the molecular dynamics of chitosan deacetylation’s effect on affinity towards collagen. The Manuscript is planned and prepared very well.
The authors address the question what type of effects of the deacetylation has on the intermolecular interactions in collagen-chitosan systems using molecular docking followed by molecular dynamics simulations. The topic is relevant to the computational chemistry of the biomaterials. The flow of the manuscript is fine and understandable to the readers.
The presented analysis will create an interest of the readers in the affinity of chitosan towards the peptides and usability of chitosan in designing for tissue engineering and biomaterials. Authors address the question satisfactorily, considering their approach in the presented methodology and computational model.
However, the conclusion can be improved a bit more to the specific to the question in rather generalizing.
Recommended for acceptance.”
Thank you for this comment. The Conclusion part was improved.
Reviewer 3 Report
Interactions between chitosan and collagen type III have been evaluated for different degrees of deacetylation and a comparison between hyaluronic acid and collagen is also given for comparison purposes. The work tries to differentiate the hydrophobic, ionic and hydrogen bonding interactions and also the role of the different amino acids. Probably, it is necessary to emphasize the interest of the analysis since the fact that the stronger hydrogen bonding interactions increased with the deacetylation degree is rather obvious. Some comments about the relevancy of the given data for subsequent investigations are needed. Some points should be considered to improve presentation and to capture the interest of the reader:
- a) Complete the introduction with experimental data concerning the variation of collagen-chitosan properties according to the deacetylation degree. Probably, some comments given in page 5 can be translated to the introduction.
- b) Try to use the same type of letters (e.g., bold, …) in equations 1 and 2.
- c) Please, expand the information corresponding to reference [56].
- d) Figure 3: Please, improve the labelling of amino acids.
- e) Figure 4b: Use the same abscise axis than for the other figures (i.e., from 0.1 to 1). Which is the result for a 100% deacetylation degree?
- f) Caption to Figure 4, line 221: “The red line” should be “The green line”. Remove “hyaluronic acid” and use only the abbreviation.
- g) Page 8, line 236: Figure 5 seems wrong (is Figure 7?).
- h) Page 8, line 247: 87.5% seems wrong since both hydrogen bonding and ionic interactions are higher for a deacetylation degree of 75%.
- i) Some points merit a more accurate discussion/justification: 1. The decrease of the H-bonds for the 60% DD. 2. The change of the trend reflecting ionic interactions at lower and higher DD of 40%. 3. The maximum ionic interactions observed at a DD of 75%.
- j) Red line has not sense in Figure 4d since it seems that there are two differentiated trends.
- k) Try to improve Figure 5 (e.g., reduce opacity) since it is difficult get any information.
- l) Lines 298 and 299 (HA) need a higher discussion in the main text.
Author Response
Dear Reviewer,
Thank you for your valuable comments, which helped us improve our paper.
We have addressed your comments as follows:
Interactions between chitosan and collagen type III have been evaluated for different degrees of deacetylation and a comparison between hyaluronic acid and collagen is also given for comparison purposes. The work tries to differentiate the hydrophobic, ionic and hydrogen bonding interactions and also the role of the different amino acids. Probably, it is necessary to emphasize the interest of the analysis since the fact that the stronger hydrogen bonding interactions increased with the deacetylation degree is rather obvious. Some comments about the relevancy of the given data for subsequent investigations are needed. Some points should be considered to improve presentation and to capture the interest of the reader:
Thank you for this general remarks. The manuscript has been improved which we hope will make it more interesting for researchers dealing with collagen blends.
- a) Complete the introduction with experimental data concerning the variation of collagen-chitosan properties according to the deacetylation degree. Probably, some comments given in page 5 can be translated to the introduction.
Thank you for this comment. It was corrected.
- b) Try to use the same type of letters (e.g., bold, …) in equations 1 and 2.
Thank you for this comment.
- c) Please, expand the information corresponding to reference [56].
Thank you. This is ref. [93] in the revised version of the manuscript. We expanded the discussion.
- d) Figure 3: Please, improve the labelling of amino acids.
Thank you for the comment. We have replaced the graphic for one with colors signifying the amino acids.
- e) Figure 4b: Use the same abscise axis than for the other figures (i.e., from 0.1 to 1). Which is the result for a 100% deacetylation degree?
Thank you. We have replaced the figure to match the range and number of points.
- f) Caption to Figure 4, line 221: “The red line” should be “The green line”. Remove “hyaluronic acid” and use only the abbreviation.
Thank you. We corrected it.
- g) Page 8, line 236: Figure 5 seems wrong (is Figure 7?).
Now it is in right order.
- h) Page 8, line 247: 87.5% seems wrong since both hydrogen bonding and ionic interactions are higher for a deacetylation degree of 75%.
Thank you. You are correct. We fixed this mistake.
- i) Some points merit a more accurate discussion/justification: 1. The decrease of the H-bonds for the 60% DD. 2. The change of the trend reflecting ionic interactions at lower and higher DD of 40%. 3. The maximum ionic interactions observed at a DD of 75%.
Thank you, the discussion was improved.
- j) Red line has not sense in Figure 4d since it seems that there are two differentiated trends.
Thank. We replaced this plot with a new one without fit line, as it shows more complex behavior. Furthermore, we removed the red line also in case of Figure 4b, since there is a lack of correlation.
- k) Try to improve Figure 5 (e.g., reduce opacity) since it is difficult get any information.
Thank you. We replaced the graphic so the surface is more visible.
- l) Lines 298 and 299 (HA) need a higher discussion in the main text.
Thank you. The Results and Discussion section was improved taking into account this issue.
Reviewer 4 Report
The authors present a molecular dynamics study towards the affinity of deacetylated chitosan towards collagen.
Introduction
While the report has merit it one major drawback is that the authors lead the reader to believe that all collagen types are equal which is simply not true. The authors need to pay more attention in the introduction section to put this work in context with regard to the types of collagen in question. While articular cartilages are 90% collagen type II there are four other distinct collagen types found in this tissue. In contrast, tendons are mostly composed of type I collagen and elastin. I point the authors to the following articles in this respect.
- 10.1034/j.1600-0838.2000.010006312.x
- 10.1016/0049-0172(91)90035-x
Furthermore, there is no consideration in the introduction to the supramolecular secondary and tertiary self-assembled and self-organised structures formed by these collagen materials in the body which is essential in putting the results into any meaningful context.
Rephrase lines 41 through 46 as the language is clumsy.
Methods
Line 94 through 99. It is unclear why the authors chose T3-785 model for human type III collagen which is a homotrimer over a model for type I which is the heterotrimer or type II which are both major component collagens.
The authors explain the methodology used for Molecular Dynamics in sufficient detail the detail on the rationale is a little thin especially the lack of discussion around the HA comparator it is unclear if the control was also modelled in the same way.
Results and Discussion
Lines 169 - 171 outlines that collagen secondary structure consists of two alpha 1 and one alpha 2 chain which is indeed representative of collagen type I however this is not the case as far as I am aware for the model collagen fragment T3-785 under consideration. Collagen type III is a homotrimer of three collagen alpha one chains as is Collagen type II.
It is recommended to revise the description of figure 2 in the text indicating to which of the varients is underdiscussion at which point.
Figure 3 is rather confusing and difficult to read. It would have been rather more simple and visually explanatory to present a single image coloured by each of the three residues than in its current form particularly as it seems that the orientation of a is completely different than that of b and c. This figure should be completely reworked or removed.
The authors state that there is a relatively weak correlation between the number of hydrogen bonds and the DD Figure 4 pannel b. Why here do the authors only present 7 rather than 8 data points as with the other panels in the figure? The authors also show a scale from 5.6 to 6.2 H bonds, could the authors reflect on their choice of parameters in relation to this result as well as what the authors are trying to indicate with fractional hydrogen bonds? With a threshold calculated surely, there should be a rather binary decision whether there is a hydrogen bond or not this would reduce their data to either 5 or 6 hydrogen bonds per degree of acetylation. I find it important that the authors try and link their calculations back to real-world observation and application in order to increase the utility of their work.
Figure 5 is again of relatively little use the authors state that the hydrogen bond distributions are presented in Figure 5 however this figure represents the electrostatic potential and the panels are so small as to show no real detail. Certainly, there is no way to verify that HYP is forming the highest number of hydrogen bonds. In fact, the figure shows no variation with 12.5% DD as is stated in the text.
Figure 6 should be redrawn to more clearly show the aspect of the electrostatic potential the authors are trying to demonstrate while the area the authors are referring to can be found with some examination it could also be interpreted that this is an overlapping part of another potential map in the z-direction of the page.
Finally, the discussion of figure 7 should be linked back to figure 4b in a more concrete manner.
Conclusions
The conclusions are relatively general with no clear statement of the context of the results. Particularly as mentioned previously the article makes no real attempt to link these findings to reality especially with the link to the 3D structures and supramolecular assemblies found in the body. The relevance of individual chain bundles to the constructions of the ECM and cartilage is rather minimal and so the utility of this work is called into question.
Apart from addressing the above points, the authors should reflect on the connection between their model and the micro-environments in which these materials will find application particularly with respect to the relevance of their choice of model.
Author Response
First of all, I would like to thank the Reviewer for his valuable comments that have contributed to a significant improvement of our manuscript. Responses to all comments are listed below. We have also changed the title to better describe the collagen type used in our study.
The authors present a molecular dynamics study towards the affinity of deacetylated chitosan towards collagen.
Introduction
While the report has merit it one major drawback is that the authors lead the reader to believe that all collagen types are equal which is simply not true. The authors need to pay more attention in the introduction section to put this work in context with regard to the types of collagen in question. While articular cartilages are 90% collagen type II there are four other distinct collagen types found in this tissue. In contrast, tendons are mostly composed of type I collagen and elastin. I point the authors to the following articles in this respect.
10.1034/j.1600-0838.2000.010006312.x
10.1016/0049-0172(91)90035-x
Furthermore, there is no consideration in the introduction to the supramolecular secondary and tertiary self-assembled and self-organised structures formed by these collagen materials in the body which is essential in putting the results into any meaningful context.
Thank you for the comment. We added more information about collagen in the manuscript.
Rephrase lines 41 through 46 as the language is clumsy.
Thank you. It was corrected.
Methods
Line 94 through 99. It is unclear why the authors chose T3-785 model for human type III collagen which is a homotrimer over a model for type I which is the heterotrimer or type II which are both major component collagens.
Thank you for this comment. You are right. However, the interactions of type III collagen with chitosan, are also of great importance in the context of osteoarthritis and tissue regeneration. We have clarified this issue in the last paragraph of the Introduction section. However, in order to be more precise, we changed the title to “Effect of chitosan deacetylation on its affinity to type III collagen: molecular dynamics study”, the purpose of the study and we corrected the entire manuscript taking into account the precise nomenclature.
The authors explain the methodology used for Molecular Dynamics in sufficient detail the detail on the rationale is a little thin especially the lack of discussion around the HA comparator it is unclear if the control was also modelled in the same way.
Thank you for this comment. We have added an explanation for the choice of collagen type III at the end of the introduction section.
We have also added a description of HA preparation. We have recalculated masses of chitosan and HA and those were around 3 kDa, not 800 Da. We apologize for the error.
Results and Discussion
Lines 169 - 171 outlines that collagen secondary structure consists of two alpha 1 and one alpha 2 chain which is indeed representative of collagen type I however this is not the case as far as I am aware for the model collagen fragment T3-785 under consideration. Collagen type III is a homotrimer of three collagen alpha one chains as is Collagen type II.
Thank you, of course, your right. We corrected this.
It is recommended to revise the description of figure 2 in the text indicating to which of the varients is underdiscussion at which point.
Thank you. The figure description was corrected.
Figure 3 is rather confusing and difficult to read. It would have been rather more simple and visually explanatory to present a single image coloured by each of the three residues than in its current form particularly as it seems that the orientation of a is completely different than that of b and c. This figure should be completely reworked or removed.
Thank you for this comment. We corrected this.
The authors state that there is a relatively weak correlation between the number of hydrogen bonds and the DD Figure 4 pannel b. Why here do the authors only present 7 rather than 8 data points as with the other panels in the figure? The authors also show a scale from 5.6 to 6.2 H bonds, could the authors reflect on their choice of parameters in relation to this result as well as what the authors are trying to indicate with fractional hydrogen bonds? With a threshold calculated surely, there should be a rather binary decision whether there is a hydrogen bond or not this would reduce their data to either 5 or 6 hydrogen bonds per degree of acetylation. I find it important that the authors try and link their calculations back to real-world observation and application in order to increase the utility of their work.
Thank you for this comment. We added the correct plot with 8 points. The previous graphic was mistakenly added before submission. With the new plot, one can see that there is no correlation between DD and the number of H-bonds. There is only a linear increase with H-bonds carried with GLY residue. The fraction indicates the average for 100 timesteps for each structure and complex for a given DD.
Figure 5 is again of relatively little use the authors state that the hydrogen bond distributions are presented in Figure 5 however this figure represents the electrostatic potential and the panels are so small as to show no real detail. Certainly, there is no way to verify that HYP is forming the highest number of hydrogen bonds. In fact, the figure shows no variation with 12.5% DD as is stated in the text.
Figure 6 should be redrawn to more clearly show the aspect of the electrostatic potential the authors are trying to demonstrate while the area the authors are referring to can be found with some examination it could also be interpreted that this is an overlapping part of another potential map in the z-direction of the page.
Thank you for these comments. Indeed, the figures were small and we corrected this. The aim of the Figures showing the electrostatic potential (Figures 6 and 7 in the revised version of the manuscript) is to show the positively charged protonated amino groups being the consequence of proton transfer. In Figure 6, only the polysaccharides were shown, since otherwise, the electrostatic potential maps graphics would be less clear for the reader. However, we showed the map of ionic contact in an enlarged manner in Figure 7. The quality of this figure was improved.
Finally, the discussion of figure 7 should be linked back to figure 4b in a more concrete manner.
Thank you for this comment. The discussion was improved (in the revised version of the manuscript the H-bonds distributions are shown in Figure 5).
Conclusions
The conclusions are relatively general with no clear statement of the context of the results. Particularly as mentioned previously the article makes no real attempt to link these findings to reality especially with the link to the 3D structures and supramolecular assemblies found in the body. The relevance of individual chain bundles to the constructions of the ECM and cartilage is rather minimal and so the utility of this work is called into question.
Apart from addressing the above points, the authors should reflect on the connection between their model and the micro-environments in which these materials will find application particularly with respect to the relevance of their choice of model.
Thank you for this comment. We extended the conclusion to give a more useful application of the results obtained in this study.
Round 2
Reviewer 1 Report
The authors made a substantial improvement to the manuscript, heeded the recommendations of the reviewers, and today they present a structured article with bibliographic support of the results obtained.
Abstract.
The authors accepted the recommendations and made the abstract more structured.
Introduction
Line 45. It is possible to supplement the information about the properties that collagen acquires when it undergoes modifications and how these can be used in the industries mentioned in line 46.
Line 82-83. The information is interesting, however, cellulose has no role in this research, so that information could be omitted and can highlight the information about the great capacity of the interaction of chitosan with other biopolymers, especially with collagen.
Conclusion.
Although it is not badly elaborated, it is suggested to reduce it in size, possibly the information concerning the final application of the possible biomaterials can be removed. I also believe that this should be framed in relation to the advantages that this study provides for researchers whose focus is the creation of the biomaterial. Well, as the authors mentioned, this article allows us to revise the experimental designs, it also helps in conducting targeted research according to the needs, and finally allows us to strengthen the databases that feed the algorithms that allow the prediction of molecular interactions.
Author Response
Dear Reviewer,
Thank you for your valuable comments, which helped us improve our paper.
We have addressed your comments as follows:
The authors made a substantial improvement to the manuscript, heeded the recommendations of the reviewers, and today they present a structured article with bibliographic support of the results obtained.
Abstract.
The authors accepted the recommendations and made the abstract more structured.
Thank you.
Introduction
Line 45. It is possible to supplement the information about the properties that collagen acquires when it undergoes modifications and how these can be used in the industries mentioned in line 46.
Thank you for the comment. We added the suggested information in the Introduction.
Line 82-83. The information is interesting, however, cellulose has no role in this research, so that information could be omitted and can highlight the information about the great capacity of the interaction of chitosan with other biopolymers, especially with collagen.
Thank you for this comment. In fact, this fragment is not directly related to the topic, so we decided to remove it. Moreover, we rebuilt the Introduction section to make it clearer and more consistent.
Conclusion.
Although it is not badly elaborated, it is suggested to reduce it in size, possibly the information concerning the final application of the possible biomaterials can be removed. I also believe that this should be framed in relation to the advantages that this study provides for researchers whose focus is the creation of the biomaterial. Well, as the authors mentioned, this article allows us to revise the experimental designs, it also helps in conducting targeted research according to the needs, and finally allows us to strengthen the databases that feed the algorithms that allow the prediction of molecular interactions.
Thank you for this comment. The Conclusion section was improved.
Reviewer 3 Report
The authors have taken into account most of the given suggestions. The paper has been improved and in my opinion, it can be published.
Author Response
Dear Reviewer
Thank you for this comment. We corrected the manuscript including the graphical issues and grammar mistakes.
Reviewer 4 Report
All points raised have been address substantially. Accept.
Author Response
Dear Reviewer,
We have corrected plots 4b and d and made a few more improvements. Thank you for reviewing our paper.